# The Nucleoid-Associated Protein Fis Represses Type 3 Fimbriae to Modulate Biofilm and Adherence Formation in *Klebsiella pneumoniae*

**DOI:** 10.3390/microorganisms13112591

**Published:** 2025-11-13

**Authors:** Santa Mejia-Ventura, Jorge Soria-Bustos, Fernando Chimal-Cázares, Gabriela Hernández-Martínez, Roberto Rosales-Reyes, Miguel A. De la Cruz, Jorge A. Yañez-Santos, Maria L. Cedillo, Gonzalo Castillo-Rojas, Dimitris Georgellis, Miguel A. Ares

**Affiliations:** 1Posgrado en Ciencias Biológicas, Facultad de Medicina, Universidad Nacional Autónoma de México, Mexico City 04510, Mexico; sljmv09@gmail.com; 2Unidad de Investigación Médica en Enfermedades Infecciosas y Parasitarias, Hospital de Pediatría, Centro Médico Nacional Siglo XXI, Instituto Mexicano del Seguro Social, Mexico City 06720, Mexico; chimal.fc@gmail.com (F.C.-C.); gabrielahmtz21@gmail.com (G.H.-M.); 3Centro de Detección Biomolecular, Benemérita Universidad Autónoma de Puebla, Puebla 72592, Mexico; jorge_s41@hotmail.com (J.S.-B.); miguel.delacruzv@correo.buap.mx (M.A.D.l.C.); jorge.yanez@correo.buap.mx (J.A.Y.-S.); lilia.cedillo@correo.buap.mx (M.L.C.); 4Unidad de Medicina Experimental, Facultad de Medicina, Universidad Nacional Autónoma de México, Mexico City 04510, Mexico; rrosalesr@ciencias.unam.mx; 5Facultad de Medicina, Benemérita Universidad Autónoma de Puebla, Puebla 72410, Mexico; 6Departamento de Microbiología y Parasitología, Facultad de Medicina, Universidad Nacional Autónoma de México, Mexico City 04510, Mexico; gcastillo55@hotmail.com; 7Departamento de Genética Molecular, Instituto de Fisiología Celular, Universidad Nacional Autónoma de México, Mexico City 04510, Mexico; dimitris@ifc.unam.mx; 8Departamento de Microbiología, Escuela Nacional de Ciencias Biológicas, Instituto Politécnico Nacional, Mexico City 11340, Mexico

**Keywords:** *Klebsiella pneumoniae*, Fis, type 3 fimbriae, *mrkA*, *mrkH*, biofilm, adherence

## Abstract

The nucleoid-associated protein Fis functions as a global regulator that influences various cellular processes in Gram-negative bacteria. In this study, we examined the role of Fis in the transcriptional regulation of type 3 fimbriae in *Klebsiella pneumoniae*, a notable opportunistic pathogen associated with hospital-acquired infections. Our transcriptional analyses revealed that deleting the *fis* gene caused a significant upregulation of *mrkA* and *mrkH*, the genes responsible for the structure and regulation of type 3 fimbriae, respectively. Additionally, phenotypic assays demonstrated that the Δ*fis* mutant exhibited enhanced biofilm formation and greater adherence to A549 lung epithelial cells compared to the wild-type strain. These effects were restored to wild-type levels in the *cis*-complemented strain. Electrophoretic mobility shift assays confirmed that Fis directly binds to the regulatory regions upstream of both *mrkA* and *mrkH*, indicating that repression occurs through direct interaction with the promoter. In summary, our findings show that Fis acts as a transcriptional repressor of *mrkA* and *mrkH*, thereby negatively regulating the expression of type 3 fimbriae, biofilm formation, and adherence. This study highlights Fis as a direct regulator of fimbrial expression and biofilm development in *K. pneumoniae*, deepening our understanding of its virulence regulatory network.

## 1. Introduction

*Klebsiella pneumoniae* is an opportunistic Gram-negative bacterium frequently found in the human gastrointestinal tract. It is recognized as one of the leading causes of hospital-acquired infections, including pneumonia, urinary tract infections, and sepsis [1,2]. Over recent decades, the rise in multidrug-resistant (MDR) and hypervirulent strains of *K. pneumoniae* has become a primary global health concern, as these pathogens display extensive antimicrobial resistance paired with increased virulence [3,4]. A key factor in the pathogenicity of *K. pneumoniae* is its impressive ability to adhere to host tissues and abiotic surfaces, forming resilient biofilms that shield the bacterium from immune clearance and antibiotic treatment [5]. Understanding the molecular mechanisms controlling adherence and biofilm formation is therefore crucial for comprehending how this pathogen persists and spreads in clinical settings.

Among the several adhesins identified in *K. pneumoniae*, type 3 fimbriae play a key role in surface attachment, biofilm maturation, and colonization of medical devices such as catheters and endotracheal tubes [6]. The structural and regulatory components of type 3 fimbriae are encoded by the *mrkABCDF* operon and its associated positive regulator, *mrkH* [7]. The major fimbrial subunit, MrkA, forms the filamentous shaft of the appendage, while MrkH acts as a c-di-GMP-dependent transcriptional activator that binds directly to the *mrkA* promoter to stimulate its expression [8]. Previous studies have shown that the nucleoid-associated protein H-NS and the global regulator IscR repress the expression of type 3 fimbriae [9,10]. Despite these findings, the role of other global or nucleoid-associated regulators in controlling fimbrial expression remains poorly understood and, in some cases, controversial, as such proteins can act as either activators or repressors depending on the genomic context and environmental conditions [11,12].

The Factor for Inversion Stimulation (Fis) is a small, homodimeric nucleoid-associated protein that plays a critical role in chromosome organization and global transcriptional regulation in Gram-negative bacteria [13,14]. Fis levels fluctuate with growth phase and nutritional status, and its DNA-bending activity can influence transcription by altering promoter topology. Depending on the genetic context, Fis can act as either a transcriptional activator or repressor, modulating various processes including virulence, motility, and metabolic adaptation [12,15,16]. Although Fis has been extensively studied in *Escherichia coli* and *Salmonella enterica*, its regulatory role in *K. pneumoniae*, particularly its potential involvement in controlling fimbrial expression and surface-associated virulence, has not been previously established.

The *fis* gene shows a notable level of sequence conservation among members of the *Enterobacteriaceae* family, highlighting its evolutionary stability and universal regulatory function. Previous studies have demonstrated that the amino acid sequences of Fis proteins are 100% identical in *K. pneumoniae*, *Serratia marcescens*, and *S*. Typhimurium [17]. Additionally, Fis proteins from *E. coli* and *K. pneumoniae* exhibit approximately 98% amino acid similarity, indicating a highly conserved structural and functional framework across these species [18].

Comparative analyses of the *fis* promoter regions in twelve bacterial species have identified a strongly conserved segment from positions −53 to +21 relative to the transcription start site of the *E. coli fis* P gene, with 99% identity observed in *K. pneumoniae* [19]. The conservation of the *fis* gene encompasses the canonical −35 and −10 promoter elements, as well as the transcription initiation site, suggesting that the mechanisms underlying *fis* expression are evolutionarily preserved across enteric bacteria.

Moreover, genome-wide analyses have uncovered potential Fis-binding motifs within the promoter regions of virulence-associated genes in *K. pneumoniae*, particularly those linked to secretion systems and surface structures [20]. Together, these findings highlight that both the coding and regulatory regions of the *fis* gene are highly conserved, reinforcing the universality of Fis-mediated transcriptional control and emphasizing the broader relevance of these results in *K. pneumoniae* and related species.

In this work, we sought to elucidate the role of the nucleoid-associated protein Fis in regulating the transcription of type 3 fimbriae in *K. pneumoniae*. Since Fis is known to be involved in global regulatory networks across *Enterobacteriaceae*, we hypothesized that it might affect the expression of key fimbrial genes related to biofilm formation and host adherence. To investigate this, we used molecular and phenotypic approaches to study the effects of *fis* deletion and complementation on fimbrial gene expression, biofilm development, and surface-associated behavior. Our findings reveal that Fis plays a role in the fine-tuning of type 3 fimbrial expression, adding a new regulatory layer to the control of adherence and biofilm formation in *K. pneumoniae*. This study contributes to a deeper understanding of how nucleoid-associated proteins regulate the expression of virulence genes in clinically important pathogens.

## 2. Materials and Methods

### 2.1. Bacterial Strains and Growth Conditions

In this study, we used the reference strain *Klebsiella pneumoniae* 123/01 [9] wild-type (WT), along with its derivative mutant Δ*fis* and the *cis*-complemented strain Δ*fis*::*fis*. A detailed description of all bacterial strains and plasmids used in this research is provided in Table 1. The cultures were routinely grown in tryptic soy broth (TSB; Difco, Beirut, Lebanon) and incubated at 37 °C under aerobic conditions. Antibiotics were added, when necessary, at the following final concentrations: [ampicillin (100 µg/mL), kanamycin (50 µg/mL), and tetracycline (10 µg/mL); Sigma-Aldrich, St. Louis, MO, USA]. For experimental assays, *K. pneumoniae* cultures were harvested during the mid-exponential growth phase, which corresponds to an optical density at 600 nm (OD_600_) of 0.6.

### 2.2. Construction of the Δfis Isogenic Mutant

The *fis* gene of *K. pneumoniae* was inactivated utilizing the λ-Red recombination system, as previously described by Datsenko and Wanner [21].In brief, a PCR fragment containing a kanamycin resistance cassette flanked by sequences homologous to the regions upstream and downstream of *fis* was amplified using gene-specific primers listed in Appendix A. The purified PCR product was then introduced into *K. pneumoniae* cells bearing the λ-Red recombinase helper plasmid pKD119 via electroporation. Recombination was stimulated by the addition of 1% (*w*/*v*) L-(+)-arabinose (Sigma-Aldrich, St. Louis, MO, USA). The successful integration of the kanamycin cassette into the *fis* locus was subsequently confirmed through PCR and sequencing.

### 2.3. Construction of the Δfis::fis Cis-Complemented Strain

The *cis*-complemented Δ*fis*::*fis* strain of *K. pneumoniae* was generated using a modified λ-Red recombination protocol based on previously described methods [22,23]. First, the FRT-flanked kanamycin resistance cassette in the Δ*fis* mutant was removed by transforming the strain with the plasmid pCP20, which expresses the FLP recombinase. Next, the *fis* coding region from *K. pneumoniae* was amplified by PCR using specific primers (Appendix A). The purified PCR product was then digested with the restriction enzymes *BamH*I and *Xho*I (New England Biolabs, Ipswich, MA, USA) and cloned into the p2795 vector. The accuracy of the plasmid construct was confirmed through DNA sequencing.

Meanwhile, a linear DNA fragment containing the *fis* gene, fused to a kanamycin resistance cassette at its 3′ end, was produced by PCR using primers with 45-nucleotide extensions homologous to the flanking regions of the chromosomal *fis* locus. This PCR product was purified with a column purification kit (Qiagen, Hilden, Germany) and introduced into *K. pneumoniae* Δ*fis*::FRT cells through electroporation. These cells also carried the λ-Red recombinase helper plasmid pKD119. Recombinase expression was induced with 1% (*w*/*v*) L-(+)-arabinose (Sigma-Aldrich, St. Louis, MO, USA). Transformants were selected on plates containing kanamycin, and their accuracy was verified by PCR and DNA sequencing to confirm proper chromosomal integration of the *fis* gene.

**Table 1 microorganisms-13-02591-t001:** Bacterial strains and plasmids used in this study.

Strains	Description *	Reference
*K. pneumoniae* WT	Wild-type *K. pneumoniae* strain 123/01, Capsular serotype, K39	[9]
*K. pneumoniae* Δ*fis*	*K. pneumoniae* Δ*fis*::FRT	This study
*K. pneumoniae* Δ*fis*::*fis*	*K. pneumoniae* Δ*fis*::*fis*-*kan* (*cis*-complemented)	This study
*E. coli* BL21 (DE3)	F^−^*omp*T *hsd*S_B_ (r_B_^−^, m_B_^−^) *gal dcm* (DE3)	Invitrogen
*E. coli* MC4100	Cloning strain F^−^ *araD139*Δ(*argF-lac*) *U169 rspL150 relA1 flbB5301 fruA25 deoC1 ptsF25*	[24]
**Plasmids**	**Description ***	**Reference**
pKD119	pINT-ts derivative containing the λ-Red recombinase system under an arabinose-inducible promoter, Tet^R^	[21]
pKD4	pANTsy derivative template plasmid containing the kanamycin cassette for λ-Red recombination, Amp^R^	[21]
pCP20	Plasmid that shows temperature-sensitive replication and thermal induction of FLP flippase synthesis, Amp^R^, Kan^R^	[21]
pMPM-T6	p15A derivative expression vector, pBAD (*ara*) promoter, Tet^R^	[25]
pT6-Fis	pMPM-T6 derivative expressing C-terminal Fis-His_6_ from pBAD (*ara*) promoter, Tet^R^	This study
p2795	pBluescript derivative template plasmid. It features a kanamycin resistance gene flanked by FRT sites; Amp^R^, Kan^R^	[22]
pKK232-9	pKK232-8 derivative vector with the promoterless *cat* reporter gene; Kan^R^	[26]
pKK-*mrkA*-*cat*	pKK232-9 derivative containing a *mrkA-cat* transcriptional fusion from nucleotides −295 to +204; Kan^R^	This study
pKK-*mrkH*-*cat*	pKK232-9 derivative containing a *mrkH-cat* transcriptional fusion from nucleotides −450 to +46; Kan^R^	This study

* Amp^R^, ampicillin resistance; Kan^R^, kanamycin resistance; Tet^R^, tetracycline resistance.

### 2.4. RNA Extraction and Reverse Transcription–Quantitative PCR Analysis

Total RNA was extracted from *K. pneumoniae* cultures collected during the mid-exponential growth phase (OD_600_ of 0.6) using the hot phenol extraction protocol described previously [27]. The RNA samples were treated with the TURBO DNA-free Kit (Invitrogen, Waltham, MA, USA) to remove any residual genomic DNA. RNA concentration and purity were measured with a NanoDrop ONE spectrophotometer (Thermo Fisher Scientific, Waltham, MA, USA), and RNA integrity was confirmed through electrophoresis on 1.5% bleach-denaturing agarose gels [28].

First-strand complementary DNA (cDNA) synthesis was performed using 1 µg of total RNA as the template with the RevertAid First Strand cDNA Synthesis Kit (Thermo Fisher Scientific, Waltham, MA, USA), following the manufacturer’s instructions. Control reactions without reverse transcriptase were included in all assays to verify the absence of genomic DNA contamination.

Reverse transcription–quantitative PCR (RT-qPCR) assays were performed using a LightCycler 480 system (Roche Diagnostics, Basel, Switzerland) along with SYBR Green I Master Mix (Roche Diagnostics, Basel, Switzerland). Each 10 µL reaction consisted of 5 µL of 2× SYBR Green I Master Mix, 2.5 µL of cDNA (approximately 25 ng), 1.5 µL of nuclease-free water, and 0.5 µL (10 µM) of each forward and reverse primer (Appendix A).

Amplifications were run in triplicate for each sample across three independent biological replicates. The *rpoD* gene served as an internal control for normalizing expression data. The amplification protocol included the following thermal profile: initial denaturation at 95 °C for 10 min; 45 cycles of denaturation at 95 °C for 10 s, annealing at 59 °C for 10 s, and extension at 72 °C for 10 s, with fluorescence acquisition at each cycle. After amplification, a melting curve analysis was performed to verify product specificity. The process began at 95 °C for 10 s, followed by a gradual increase from 65 °C to 97 °C with continuous fluorescence detection, and concluded with a cooling phase at 40 °C for 10 s. To ensure analysis accuracy, control reactions lacking cDNA template or reverse transcriptase were systematically included. Gene expression differences among the WT, Δ*fis*, and Δ*fis*::*fis* strains were quantified using the 2^−ΔΔCt^ method [29,30], facilitating reliable comparisons of transcriptional changes associated with Fis’s regulation in *K. pneumoniae*. Results reports represent the mean of three independent biological replicates.

### 2.5. Construction of Transcriptional Reporter Fusions

Promoter regions corresponding to the *mrkA* and *mrkH* genes of *K. pneumoniae* were amplified by PCR using specific primers designed to include restriction sites for *BamH*I and *Hind*III. The amplified fragments were then digested with the same enzymes (New England Biolabs, Ipswich, MA, USA) and ligated into the pKK232-9 vector [26], which contains a promoterless *cat* gene that encodes chloramphenicol acetyltransferase. The ligation reactions were performed using T4 DNA ligase (Thermo Fisher Scientific, Waltham, MA, USA) according to the manufacturer’s instructions.

Recombinant plasmids were verified through restriction analysis and sequencing to ensure the correct orientation and integrity of each promoter insert. The confirmed constructs were subsequently introduced into the corresponding *K. pneumoniae* strains by electroporation under standard conditions.

### 2.6. Determination of Chloramphenicol Acetyltransferase Activity

Chloramphenicol acetyltransferase (CAT) activity was determined using a continuous colorimetric assay based on 5,5′-dithiobis-(2-nitrobenzoic acid) (DTNB), adapted for a microplate format [31,32]. One milliliter of cultures grown to mid-logarithmic phase (OD600 = 0.6) was harvested by centrifugation at 16,000× *g* for 10 min at 4 °C. Cell pellets were washed with 1 mL of TDTT buffer (50 mM Tris–HCl, pH 7.8; 30 µM DTT [dithiothreitol]; Sigma-Aldrich, St. Louis, MO, USA) and resuspended in 500 µL of the same buffer. Cells were disrupted by sonication on ice using 10 s pulses alternated with cooling intervals until the suspension became translucent, indicating complete cell lysis. Cell debris was removed by centrifugation at 16,000× *g* for 15 min at 4 °C, and clarified supernatants were collected in clean microtubes and kept on ice for immediate analysis.

CAT assays were conducted in flat-bottom 96-well microplates (Corning Costar, New York, NY, USA) by adding 10 µL of cell extract to 200 µL of reaction buffer containing 1 mM DTNB (Sigma-Aldrich, St. Louis, MO, USA), 0.1 mM acetyl-CoA (Sigma-Aldrich, St. Louis, MO, USA), and 0.1 mM chloramphenicol (Sigma-Aldrich, St. Louis, MO, USA) in 0.1 M Tris–HCl, pH 7.8 (Sigma-Aldrich, St. Louis, MO, USA). The acetylation of chloramphenicol catalyzed by CAT releases CoA-SH, which reacts stoichiometrically with DTNB to produce 2-nitro-5-thiobenzoic acid (TNB), a yellow anion with strong absorbance at 412 nm [33]. Absorbance at 412 nm was monitored kinetically every second for 2 min at 25 °C using a Varioskan LUX multimode microplate reader and its integrated Skanlt software version 6.1 (Thermo Fisher Scientific, Waltham, MA, USA) in kinetic mode. Blanks without enzyme extract were included to correct for non-enzymatic reactions. Additional negative controls lacking acetyl-CoA or chloramphenicol were included to detect any non-specific reactions or background activity.

Enzymatic activity was determined from the initial linear slope (ΔA/Δt, in Abs/min) after blank correction. The slope was converted to the amount of product using the molar extinction coefficient of TNB (ε412 = 13,600 M^−1^·cm^−1^) and an effective path length (*ℓ*) of 0.56 cm, corresponding to a 210 µL volume in microplate wells, based on Ellman’s method [33] and bacterial CAT assay adaptations [31]. The reaction rate was calculated as$$Rate\left(\mu mol/min\right)=\frac{\Delta A/\Delta t}{\varepsilon \cdot l}\times V\times 10^{6}$$

ΔA/Δt is the slope of the linear portion of the reaction curve (Abs/min), ε is the molar extinction coefficient (M^−1^·cm^−1^), *ℓ* is the optical path length (cm), and V is the total reaction volume (L). Specific activity was expressed as µmol of chloramphenicol acetylated per minute per mg of total protein:$$Specific\;activity(\mu mol/min/mg)=\frac{Rate(\mu mol/min)}{Protein(mg)}$$

Total protein concentration in cell extracts was measured using the Bradford method (Bradford Reagent; Bio-Rad Laboratories, Hercules, CA, USA) with bovine serum albumin (BSA; Sigma-Aldrich, St. Louis, MO, USA) as the standard [34]. Only linear time intervals with coefficients of determination (R^2^) ≥ 0.98 were used for activity calculations. Reported values are the mean of two independent biological replicates.

### 2.7. Quantitative Assessment of Biofilm Formation on Polystyrene Surfaces

The ability of bacterial cells to adhere and form biofilms on abiotic surfaces was evaluated using sterile, flat-bottom 96-well polystyrene microplates (Costar, Corning Inc., Corning, NY, USA), following a modified version of the method described by Saldaña et al. [35]. Overnight cultures grown in TSB were diluted 1:100 in fresh sterile TSB medium. Aliquots of 100 µL of the resulting suspension were dispensed into individual wells (performed in triplicate), and the plates were incubated under static conditions at room temperature (approximately 25 °C) for 24 h to allow biofilm formation.

After incubation, planktonic cells and non-adherent bacteria were gently removed by aspiration, and each well was washed three times with sterile phosphate-buffered saline (PBS, pH 7.4) to remove any residual unattached cells. The attached biofilm biomass was stained by adding 1% (*w*/*v*) crystal violet (CV) solution to each well and incubating for 20 min at room temperature in the dark. The excess stain was carefully discarded, and the wells were washed three times with PBS to eliminate any unbound dye. Subsequently, the bound CV was solubilized by adding 100 µL of 70% (*v*/*v*) ethanol to each well and incubating for 15 min with gentle shaking to ensure complete dye extraction.

Quantification of biofilm formation was conducted by measuring the optical density (OD) of the extracted dye at 600 nm utilizing a Multiskan microplate reader (Thermo Fisher Scientific, Waltham, MA, USA). The mean absorbance values were calculated from three biological replicates per condition. Negative controls containing only sterile TSB were included to account for background absorbance.

### 2.8. Quantitative Adherence Assay to Cultured Eukaryotic Cells

The adherence of bacteria to eukaryotic cells was tested using monolayers of the human alveolar epithelial cell line A549 (ATCC CCL-185). Briefly, confluent monolayers containing approximately 7 × 10^5^ cells per well were prepared in 24-well tissue culture plates and maintained in Dulbecco’s Modified Eagle Medium (DMEM; Invitrogen, Carlsbad, CA, USA) supplemented with 10% (*v*/*v*) fetal bovine serum (FBS; Gibco, Thermo Fisher Scientific, Waltham, MA, USA) and 1% penicillin–streptomycin (Sigma-Aldrich, St. Louis, MO, USA) at 37 °C in a humidified atmosphere with 5% CO_2_.

Overnight bacterial cultures grown in TSB were adjusted to an optical density that corresponds to a multiplicity of infection (MOI) of 100 and added to the cell monolayers in antibiotic-free DMEM. Infections were allowed to proceed for 2 h at 37 °C under 5% CO_2_ to facilitate bacterial attachment. After incubation, the medium was carefully aspirated, and the wells were washed three times with sterile phosphate-buffered saline (PBS, pH 7.4) to remove non-adherent bacteria.

To quantify adherent bacteria, epithelial cells were lysed by adding 0.1% (*v*/*v*) Triton X-100 (Sigma-Aldrich, St. Louis, MO, USA) in PBS and incubated for 10 min at room temperature with gentle agitation to ensure thorough cell disruption. The lysates were homogenized, serially diluted tenfold in sterile PBS, and plated on LB agar plates for enumeration of viable bacterial colony-forming units (CFUs). Each assay was performed in triplicate and independently repeated at least three times on different days. Data are shown as the mean CFU/mL.

### 2.9. Purification of Recombinant Fis-His_6_ Protein

The recombinant Fis protein, fused to a C-terminal hexahistidine tag (Fis-His_6_), was expressed in *E. coli* BL21 (DE3) cells and purified under denaturing conditions using Ni-NTA affinity chromatography. The expression plasmid pT6-Fis was introduced into electrocompetent *E. coli* BL21 (DE3) cells via electroporation, and transformants were selected on LB agar supplemented with the appropriate antibiotic. Single colonies were inoculated into LB broth and grown at 37 °C with shaking until the cultures reached mid-exponential phase (OD_600_ of 0.6). Recombinant protein expression was induced by adding 1% (*w*/*v*) L(+)-arabinose (Sigma-Aldrich, St. Louis, MO, USA) and incubating for 6 h at 37 °C. Cells were harvested by centrifugation at 12,000× *g* for 10 min at 4 °C and resuspended in a solubilization buffer containing 8 M urea, 100 mM Na_2_HPO_4_, and 10 mM Tris-HCl (pH 8.0). 

Cell disruption was achieved by sonication on ice, applying 10 s pulses alternated with 10 s cooling intervals until the suspension turned translucent, indicating complete lysis. Insoluble debris was removed through centrifugation at 16,000× *g* for 20 min at 4 °C. The clarified supernatant was then filtered through a 0.45 µm membrane before being loaded onto a pre-equilibrated Ni-NTA agarose column (Qiagen, Hilden, Germany). Bound proteins were washed with 200 mL of washing buffer (50 mM imidazole, 10 mM Na_2_HPO_4_, 1.8 mM KH_2_PO_4_, 137 mM NaCl, and 2.7 mM KCl; pH 7.4) to remove nonspecifically bound contaminants. The Fis-His_6_ protein was eluted using 10 mL of elution buffer containing 500 mM imidazole in the same phosphate buffer system. 

Eluted fractions were analyzed by sodium dodecyl sulfate–polyacrylamide gel electrophoresis (SDS-PAGE 12%) followed by Coomassie Brilliant Blue R-250 staining to evaluate purity. The pooled elution fractions were dialyzed at 4 °C using a dialysis tubing cellulose membrane (Sigma-Aldrich, St. Louis, MO, USA) to remove imidazole and facilitate the refolding of the Fis-His_6_ recombinant protein. We performed the dialysis by using 1 L of renaturation buffer (20 mM Tris-HCl (pH 7.5), 150 mM KCl, 5 mM MgCl_2_, 1 mM DTT, and 10% (*v*/*v*) glycerol; Sigma-Aldrich, St. Louis, MO, USA). Over a period of 12 h, we carried out three buffer changes. This buffer composition was optimized to replicate near-physiological ionic strength and redox conditions, which are essential for the proper folding and stabilization of the Fis homodimer. The presence of Mg^2+^ and glycerol contributed to structural stabilization and minimized aggregation during the renaturation process.

After dialysis, the sample was centrifuged at 12,000× *g* for 10 min at 4 °C to remove insoluble material. The supernatant containing soluble, renatured Fis-His_6_ was collected, and protein concentration was measured using the Bradford colorimetric assay (Bio-Rad Protein Assay, Hercules, CA, USA) with bovine serum albumin (BSA) as the standard. The purified Fis-His_6_ protein was aliquoted and stored at −70 °C in a stabilization buffer (50% (*v*/*v*) glycerol, 10 mM Na_2_HPO_4_, 1.8 mM KH_2_PO_4_, 137 mM NaCl, and 2.7 mM KCl (pH 7.4); Sigma-Aldrich, St. Louis, MO, USA) to preserve its structural integrity and functional activity during long-term storage.

### 2.10. Electrophoretic Mobility Shift Assays

Electrophoretic mobility shift assays (EMSA) were performed to examine the binding interactions of the recombinant Fis-His_6_ protein with the regulatory regions of the *mrkA* and *mrkH* genes in *K. pneumoniae*. A 502 bp DNA fragment corresponding to the regulatory region upstream of *mrkA* and a 500 bp fragment representing the *mrkH* regulatory region were amplified by PCR using genomic DNA from *K. pneumoniae* strain 123/01 as the template. The purified DNA probes were adjusted to a final concentration of 100 ng per reaction.

Protein–DNA binding reactions were carried out in a total volume of 20 µL using a 1× H/S binding buffer (40 mM HEPES, 8.0 mM MgCl_2_, 50 mM KCl, 1.0 mM DTT, 0.05% NP-40, and 0.1 mg/mL BSA (Sigma-Aldrich, St. Louis, MO, USA)). Increasing concentrations of purified recombinant Fis-His_6_ protein (0.0–0.8 µM) were incubated with each DNA probe for 20 min at room temperature to allow equilibrium complex formation. To assess binding specificity, a non-specific control probe (100 ng) corresponding to a coding fragment of the *Mycobacterium tuberculosis fbpA* gene was included in the same reactions.

Samples were resolved by electrophoresis on 6% non-denaturing polyacrylamide gels prepared in 0.5× Tris–borate–EDTA (TBE; Invitrogen, Waltham, MA, USA) buffer. Gels were run at a steady voltage of 120 V under controlled temperature conditions to prevent dissociation of the protein–DNA complexes. After electrophoresis, the gels were stained with ethidium bromide (0.5 µg/mL) for 10 min, quickly rinsed with distilled water, and visualized under UV transillumination using a BioDoc-It Imaging System (UVP Analytik Jena, Upland, CA, USA).

The appearance of retarded (shifted) bands relative to the free DNA probe indicated the formation of specific Fis–DNA complexes. The relative intensity of these shifted bands was used to estimate the binding affinity of Fis for the *mrkA* and *mrkH* regulatory regions, providing evidence of its potential role in directly modulating type 3 fimbrial gene expression in *K. pneumoniae*.

### 2.11. Identification of Putative Fis-Binding Motifs in the mrkA and mrkH Regulatory Regions

Putative Fis-binding motifs were identified in the regulatory regions of the *mrkA* and *mrkH* genes of *K. pneumoniae* using the web-based database and analysis platform, PRODORIC v2.0 (https://www.prodoric.de/ (accessed on 18 September 2025)) [36]. For each gene, we retrieved a 500-nucleotide sequence located upstream of the translational start codon (ATG) for the *mrkA* and *mrkH* genes and analyzed it for the presence of potential Fis-binding sites. We utilized the Virtual Footprint tool, a component of the PRODORIC suite, to conduct the motif search and comparative analysis. The predictions were based on the Fis consensus recognition sequence 5′-GNNNAWWWWWTNNNC-3′, derived from experimentally validated Fis-binding sites in *E. coli* [37]. Default search parameters were applied, allowing for up to two mismatches relative to the canonical motif to account for sequence variability among orthologous promoters.

### 2.12. Statistical Analyses

Statistical analyses were conducted using GraphPad Prism version 10.5.0 (GraphPad Software, San Diego, CA, USA). Data were analyzed using one-way analysis of variance (ANOVA), followed by Tukey’s post hoc test to assess differences among groups. A *p*-value of less than 0.05 was considered statistically significant.

## 3. Results

### 3.1. Fis Is Required for Optimal Growth of K. pneumoniae

We monitored the growth dynamics of *K. pneumoniae* WT, Δ*fis*, and Δ*fis*::*fis* strains in LB broth over 8 h at 37 °C. The Δ*fis* mutant showed reduced growth compared to the WT strain, with a significant decrease during the exponential phase and a moderate decline in the stationary phase (Figure 1).

In contrast, the Δ*fis*::*fis* strain restored growth kinetics to levels like the WT, confirming that the absence of Fis specifically caused the growth defect. These results suggest that Fis is essential for optimal bacterial proliferation in *K. pneumoniae*, consistent with its known role in promoting growth and global transcriptional activity in other enterobacteria. Fis, as shown in previous studies, accumulates during the early exponential phase and promotes the expression of genes involved in energy metabolism and DNA topology, which are critical processes supporting rapid cell growth [13,38]. Therefore, the slower growth rate observed in the Δ*fis* mutant likely reflects impaired activation of these global growth-related pathways.

### 3.2. Fis Negatively Modulates the Expression of Type 3 Fimbrial Genes mrkA and mrkH

To determine the role of Fis in regulating type 3 fimbrial expression, we analyzed the transcription levels of *mrkA* and *mrkH* in WT, Δ*fis*, and Δ*fis*::*fis* strains of *K. pneumoniae*. RT-qPCR analysis revealed that the deletion of *fis* resulted in a significant upregulation of both genes, indicating that Fis exerts a repressive influence on their transcription. The transcription levels of *mrkA* and *mrkH* increased 8.6-fold and 9.7-fold, respectively, compared to the WT strain, demonstrating that Fis strongly represses the expression of these fimbrial components. In the complemented strain, transcription levels returned to those similar to the WT, confirming that the observed effects were specifically due to the absence of Fis (Figure 2A,B).

To support these findings, we transcriptionally fused the promoter regions of *mrkA* and *mrkH* to the *cat* reporter gene, which encodes chloramphenicol acetyltransferase (CAT). These reporter constructs were introduced into the WT, Δ*fis*, and Δ*fis*::*fis* backgrounds, and CAT activity was measured as an indicator of promoter strength. Consistent with the RT-qPCR data, both promoter fusions exhibited significantly higher CAT activity in the Δ*fis* mutant, 11.8-fold for *mrkA* and 9.6-fold for *mrkH*, compared to the WT strain, reflecting increased transcriptional activity in the absence of Fis (Figure 2C,D).

The restoration of Fis expression in the complemented strain reduced CAT activity to WT levels, reinforcing the idea that Fis acts as a transcriptional repressor of *mrkA* and *mrkH*. Together, these results identify Fis as a key modulator of type 3 fimbrial gene expression in *K. pneumoniae*, providing genetic and functional evidence of its repressive role. Since *mrkA* and *mrkH* encode essential components for fimbrial assembly and regulation, the significant upregulation of their transcription in the absence of Fis is expected to influence phenotypes associated with surface attachment. Given that type 3 fimbriae play a critical role in biofilm formation and adherence to epithelial cells, we next investigated whether the loss of *fis* affects these virulence-related traits in *K. pneumoniae*.

### 3.3. Fis Suppresses Biofilm Formation and Host–Cell Adherence

To investigate whether the absence of Fis affects surface-associated characteristics, we examined biofilm formation and adherence in *K. pneumoniae* WT, Δ*fis*, and Δ*fis*::*fis* strains. Quantitative biofilm assays conducted in polystyrene microtiter plates showed that deletion of the *fis* gene significantly increased biofilm biomass compared to the WT strain, indicating that Fis negatively regulates biofilm development (Figure 3A,B). This observation aligns with previous reports identifying type 3 fimbriae as major contributors to biofilm formation in *K. pneumoniae* [39]. In contrast, the complemented strain restored biofilm levels to those of the WT, confirming that the absence of Fis specifically caused the hyper-biofilm phenotype.

In summary, these results demonstrate that Fis plays a crucial role in regulating surface-associated virulence traits in *K. pneumoniae*. By repressing the expression of the type 3 fimbrial genes *mrkA* and *mrkH*, Fis limits biofilm formation and host–cell adherence. This regulation is essential for maintaining the balance between the sessile and planktonic lifestyles of the bacterium on both abiotic and biotic surfaces.

### 3.4. Fis Directly Interacts with the mrkA and mrkH Promoter Region

Electrophoretic mobility shift assays demonstrated that Fis directly binds to the promoter regions of the *mrkA* and *mrkH* genes. A distinct, concentration-dependent retardation in DNA mobility was observed for both promoters, reflecting the formation of stable Fis–DNA complexes and confirming that Fis specifically associates with the regulatory regions controlling type 3 fimbrial expression (Figure 4A,B). The appearance of shifted bands at increasing protein concentrations is consistent with the presence of more than one Fis-binding site within each promoter fragment.

To further validate the specificity of this interaction, a control probe containing the promoter region of the *fbpA* gene, which is unrelated to type 3 fimbrial regulation, was analyzed in parallel under identical conditions. No mobility shift was detected for this fragment, supporting that Fis selectively binds to the *mrkA* and *mrkH* promoters and not to unrelated genomic sequences (Figure 4A,B). This selective binding pattern aligns with the transcriptional data, which show derepression of *mrkA* and *mrkH* in the Δ*fis* mutant, reinforcing the conclusion that Fis acts as a direct repressor of these genes.

Together, these results establish Fis as a DNA-binding repressor that specifically recognizes the *mrkA* and *mrkH* promoters, thereby regulating type 3 fimbrial expression and affecting surface-associated virulence traits, such as biofilm formation and adherence, in *K. pneumoniae*.

### 3.5. Fis-Binding Motifs Identified in the Regulatory Regions of mrkA and mrkH

To further investigate the regulatory role of Fis, we conducted an in silico analysis of the upstream regions of the *mrkA* and *mrkH* genes to identify potential Fis-binding sites. The regulatory region of *mrkA* contained three predicted Fis-binding motifs, all located downstream of the previously reported transcription start site (TSS, +1) [8] (Figure 5A).

Interestingly, the promoter region of *mrkH* had three putative Fis-binding motifs, with two positioned upstream and one downstream of its experimentally defined TSS [40] (Figure 5B). The distribution of these motifs suggests that Fis may repress transcription by binding within or near the core promoter regions, thereby hindering RNA polymerase access.

Comparative sequence alignment using the *E. coli* Fis consensus motif (5′-GNNNAWWWWWTNNNC-3′) [37] showed a high level of conservation among the predicted sites. The motifs identified within the *mrkA* and *mrkH* promoters shared 13 to 14 matching nucleotides out of 15 with the *E. coli* consensus sequence, suggesting these are likely genuine Fis-binding sites (Figure 5C). 

Overall, these findings support the idea that Fis directly interacts with the promoter regions of *mrkA* and *mrkH* through conserved recognition motifs. This spatial arrangement of binding sites aligns with the observed transcriptional repression of both genes, highlighting Fis as a key regulator that modulates type 3 fimbrial expression in *K. pneumoniae*.

## 4. Discussion

This study demonstrates that the nucleoid-associated protein Fis acts as a transcriptional repressor of type 3 fimbrial expression in *K. pneumoniae*, influencing two major virulence-related phenotypes: biofilm formation and epithelial adherence. Through a combination of transcriptional, phenotypic, and in silico analyses, we provide evidence that Fis directly binds to the promoter regions of *mrkA* and *mrkH*, repressing their expression and thereby modulating the transition between planktonic and surface-associated lifestyles.

Fis is a global regulator known to influence a wide range of cellular functions, including DNA topology, replication, and transcription in Gram-negative bacteria [13,38,41,42]. Its expression is tightly growth-phase dependent, peaking during the early exponential phase when cells are actively dividing and decreasing as cultures enter the stationary phase [42,43]. In *E. coli* and *S. enterica*, Fis positively regulates genes associated with tRNA transcription and central metabolism, thereby promoting rapid proliferation [13,42]. Consistent with this regulatory role, our data show that *K. pneumoniae* cells lacking Fis exhibit reduced growth during the exponential phase, underscoring the conserved function of this regulator in optimizing bacterial physiology during nutrient-rich conditions.

Beyond its role in growth, Fis has been increasingly recognized as a central modulator of virulence gene expression across *Enterobacteriaceae*. In *Salmonella*, Fis activates genes of the type III secretion system (T3SS) and contributes to intestinal colonization [42]. In contrast, in *E. coli*, it modulates the expression of flagellar and fimbrial operons in coordination with other global regulators [16]. This study enhances our understanding of *K. pneumoniae* by identifying Fis as a direct repressor of type 3 fimbrial genes. Our results show that Fis binds to multiple sites within the *mrkA* and *mrkH* promoters, which decreases their transcription. This finding is supported by both RT-qPCR and promoter–reporter fusion analyses. Electrophoretic mobility shift assays using probes that cover the entire regulatory regions of *mrkA* and *mrkH* demonstrated a specific interaction between purified Fis and the promoter DNA, providing experimental evidence of direct binding.

In addition, subsequent in silico analysis identified three potential Fis-binding motifs that align with the *E. coli* consensus sequence. These motifs are located within functionally important regions of the promoter, with one overlapping the predicted *mrkH* –35 box. This suggests that Fis binding could influence promoter accessibility, promoter clearance, or RNA polymerase elongation, which is a mechanism previously proposed for other Fis-repressed genes in *E. coli* [15].

Overall, these findings provide strong evidence for a direct regulatory role of Fis in controlling the expression of type 3 fimbriae in *K. pneumoniae*. Future work in our laboratory will focus on experimentally validating these in silico predictions through targeted mutagenesis and complementary biochemical analyses to gain a more definitive understanding of this regulatory mechanism.

The regulatory balance between Fis and positive activators such as MrkH is likely essential for fine-tuning fimbrial expression in response to environmental cues. MrkH, a c-di-GMP-dependent transcriptional activator, directly stimulates *mrkA* transcription, promoting biofilm formation [8]. Our findings suggest that Fis antagonizes this activation by binding to overlapping or adjacent promoter regions, thereby ensuring that fimbrial expression is tightly controlled. Such dual regulatory systems are standard among nucleoid-associated proteins, where dynamic interplay between repressors and activators allows bacteria to modulate surface structures according to physiological status or environmental stress [38].

The influence of Fis on biofilm formation and adherence is consistent with its broader role in regulating surface-associated traits globally. The Δ*fis* mutant exhibited a hyper-biofilm phenotype and enhanced adherence to A549 epithelial cells, both of which correlated with increased *mrkA* and *mrkH* expression. This phenotype parallels reports in other bacterial systems, where the loss of Fis leads to the overexpression of adhesive or stationary-phase genes, such as curli and type 1 fimbriae in *E. coli*, and the T3SS in *Salmonella* [42,44,45]. By repressing type 3 fimbrial expression, Fis may help maintain a balance between motile and sessile states, preventing excessive surface adhesion during rapid growth phases and allowing for a more coordinated virulence response.

From a broader perspective, the function of Fis in *K. pneumoniae* appears to integrate into a complex network of nucleoid-associated proteins that collectively shape virulence gene expression. For instance, H-NS, another major chromatin-associated protein, represses *mrk* gene transcription by silencing AT-rich regions within the promoter [9]. At the same time, IscR and CRP modulate fimbrial expression in response to metabolic and environmental signals [10,46]. Our data suggest that Fis operates within this network as a negative regulator, potentially counteracting the activity of MrkH or other transcriptional activators to ensure that type 3 fimbrial expression occurs only under specific physiological conditions. The presence of conserved Fis-binding motifs in both *mrkA* and *mrkH* promoters, closely resembling those characterized in *E. coli* [47], supports the idea of a conserved DNA recognition mechanism shared among *Enterobacteriaceae*.

Our findings identify a new layer of transcriptional regulation that controls type 3 fimbriae expression in *K. pneumoniae*. The protein Fis directly represses the genes *mrkA* and *mrkH*, fine-tuning adhesin expression essential for biofilm formation and host colonization. This research broadens the understanding of Fis, highlighting its role in modulating virulence.

In addition to regulating the *mrk* genes, Fis may also affect other virulence factors in *K. pneumoniae*, such as capsule biosynthesis, siderophore production, and modifications to lipopolysaccharide (LPS). Although this study did not directly examine these factors, we encourage further exploration of these mechanisms in future research to demonstrate the extensive regulatory potential of Fis and suggest that its role in the virulence of *K. pneumoniae* extends beyond just the regulation of fimbriae, as in other bacteria, such as in *S. enterica* [48].

Another intriguing possibility is that Fis may contribute to antibiotic resistance in *K. pneumoniae*. In *Pseudomonas aeruginosa*, disruption of the *fis* gene increases susceptibility to ciprofloxacin, suggesting that Fis plays a role in maintaining resistance and modulating stress-related pathways, including pyocin biosynthesis [49]. While its direct association with antibiotic resistance in *K. pneumoniae* remains to be confirmed, it is plausible that Fis influences efflux pump activity, membrane physiology, or other mechanisms related to stress adaptation that may modulate antimicrobial susceptibility [50,51]. This hypothesis represents an important direction for future research on the global regulatory network controlled by Fis.

Given that Fis controls genes related to adhesion and surface structures, it is reasonable to speculate that Fis may also influence biofilm formation in polymicrobial environments. Mixed-species biofilms are increasingly recognized as significant in both clinical and environmental contexts [52]. The modulation of adhesion factors in *K. pneumoniae* by Fis could impact its interactions with other microorganisms. Future studies that examine the function of Fis in multispecies biofilm models will help clarify its role in community behavior and ecological fitness.

## 5. Conclusions

This study identifies the nucleoid-associated protein Fis as a key regulator that represses type 3 fimbrial genes in *K. pneumoniae*. Our analyses show that Fis binds to the promoter regions of *mrkA* and *mrkH*, leading to their repression. Loss of the *fis* gene results in increased expression of these genes, enhanced biofilm formation, and greater adherence to epithelial cells, with these effects reversed upon complementation. These findings highlight the role of Fis in regulating the balance between the attached and free-floating lifestyles of *K. pneumoniae* by controlling fimbrial expression, adherence, and biofilm development. Overall, the study provides insights into the mechanisms governing virulence and identifies Fis as a potential target for strategies to combat biofilm-associated infections caused by *K. pneumoniae*.

## Figures and Tables

**Figure 1 microorganisms-13-02591-f001:**
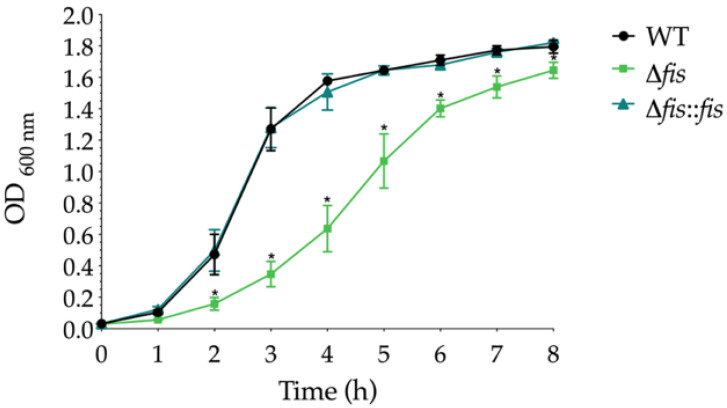
Growth curves of *K. pneumoniae* WT, Δ*fis*, and Δ*fis*::*fis* strains. Cultures were grown in TSB at 37 °C for 8 h, and the optical density at 600 nm (OD_600_) was measured at hourly intervals. Data represent the mean ± standard deviation from three independent biological replicates. Statistical significance was determined using one-way ANOVA followed by Tukey’s post hoc test, comparing to the growth of the WT strain (* *p* < 0.05).

**Figure 2 microorganisms-13-02591-f002:**
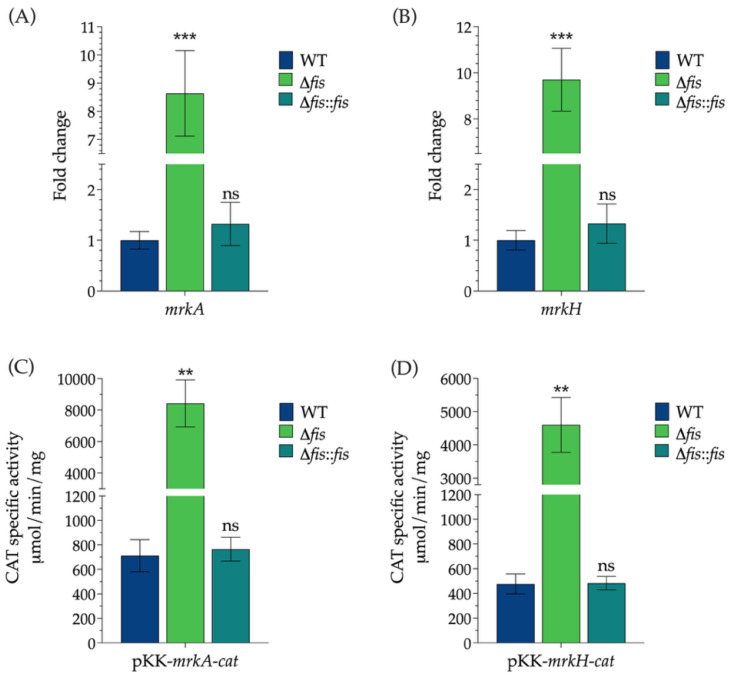
Regulation of *mrkA* and *mrkH* type 3 fimbrial gene expression by the nucleoid-associated protein Fis. RT-qPCR analysis of (**A**) *mrkA* and (**B**) *mrkH* expression, and measurement of CAT activity from transcriptional fusions to the *cat* reporter gene (**C**) pKK-*mrkA*-*cat* and (**D**) pKK-*mrkH*-cat in *K. pneumoniae* WT, Δ*fis*, and Δ*fis*::*fis* strains. Data represent the mean values ± standard deviation from at least two independent biological replicates. Statistical significance was determined relative to the WT strain using one-way ANOVA followed by Tukey’s post hoc test (** *p* < 0.01; *** *p* < 0.001; ns, not significant).

**Figure 3 microorganisms-13-02591-f003:**
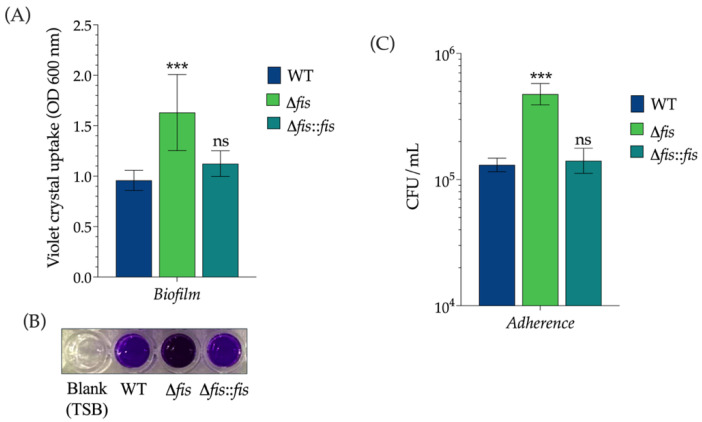
Effect of Fis on biofilm formation and adherence to epithelial cells in *K. pneumoniae*. (**A**) Quantitative biofilm assay performed in polystyrene microtiter plates showing biofilm biomass of WT, Δ*fis*, and Δ*fis*::*fis* strains. (**B**) Representative crystal violet-stained biofilms formed by the indicated strains in 96-well plates. The blank corresponds to TSB medium without bacterial inoculation and was included as a negative control. (**C**) Adherence of the same strains to A549 lung epithelial cells. The Δ*fis* mutant exhibited significantly increased biofilm formation and adherence compared with the WT strain, whereas complementation (Δ*fis*::*fis*) restored these phenotypes to WT levels. Data represent mean values ± standard deviation from three independent biological experiments. Statistical significance was determined relative to the WT strain using one-way ANOVA followed by Tukey’s post hoc test (*** *p* < 0.001; ns, not significant).

**Figure 4 microorganisms-13-02591-f004:**
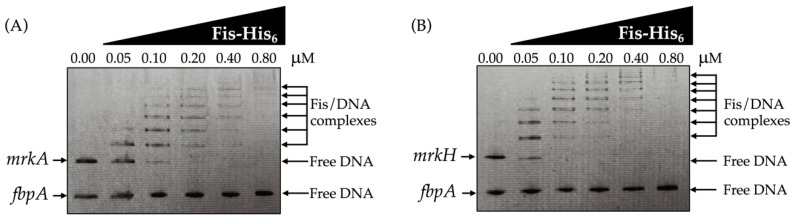
Direct interaction of Fis with the *mrkA* and *mrkH* promoter regions was demonstrated by electrophoretic mobility shift assays (EMSA). Increasing concentrations of purified Fis-His_6_ recombinant protein were incubated with DNA fragments containing the promoter regions of (**A**) *mrkA* and (**B**) *mrkH*. A progressive, concentration-dependent retardation in DNA mobility was observed for both fragments, indicating the formation of Fis–DNA complexes. As a negative control, a DNA fragment corresponding to the *fbpA* coding region, unrelated to type 3 fimbrial regulation, was included in the same assays, showing no detectable shift. Arrows indicate free DNA, or Fis/DNA complexes, stained with ethidium bromide.

**Figure 5 microorganisms-13-02591-f005:**
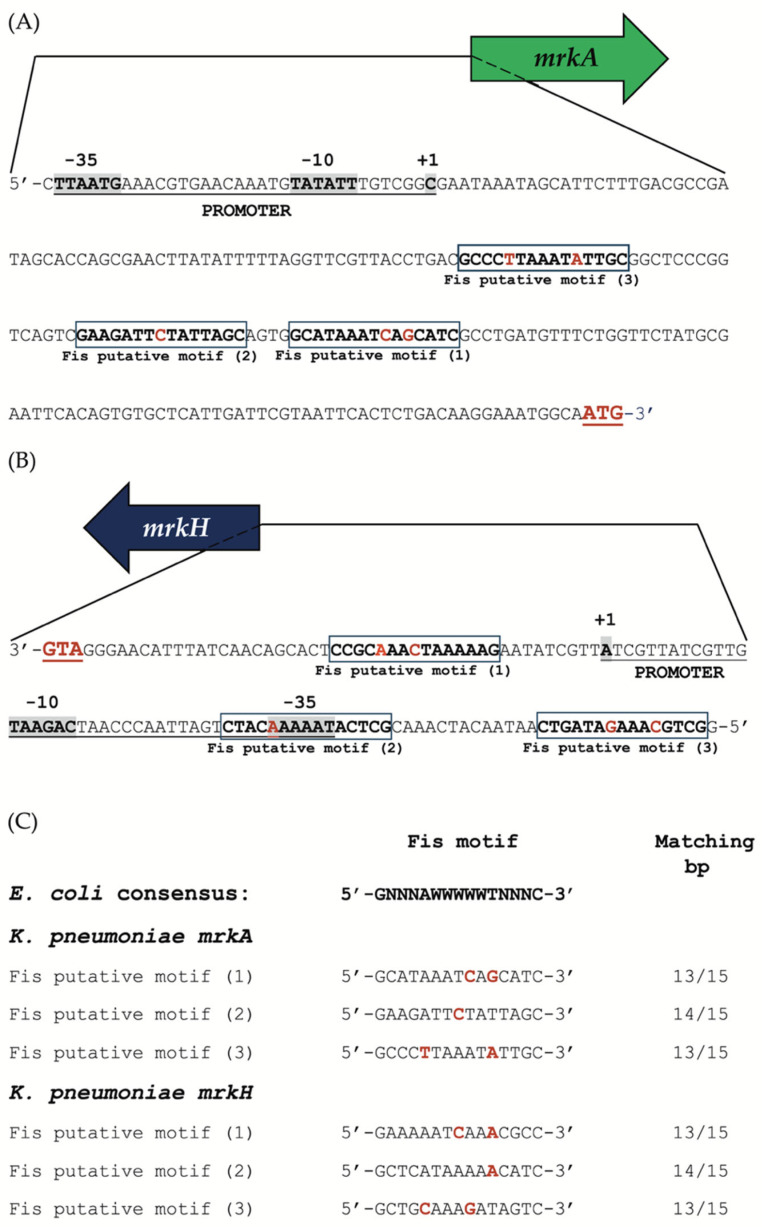
In silico analysis of the regulatory regions of *mrkA* and *mrkH*. (**A**) Predicted Fis-binding motifs identified within the upstream regulatory region of *mrkA*. (**B**) Predicted Fis-binding motifs identified within the upstream regulatory region of *mrkH*. Putative Fis-binding sites are enclosed in boxes; nucleotides matching the consensus sequence are shown in bold, whereas non-conserved positions are indicated in bold red. The experimentally determined promoter regions for each gene are also shown. (**C**) Alignment of the predicted Fis-binding motifs from the *K. pneumoniae mrkA* and *mrkH* regulatory regions with the consensus Fis-binding sequence from *E. coli*. Nucleotides that differ from the *E. coli* consensus are highlighted in bold red.

## Data Availability

The original contributions presented in the study are included in the article/Appendix A, further inquiries can be directed to the corresponding author.

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
