# Peer review of "The Nucleoid-Associated Protein Fis Represses Type 3 Fimbriae to Modulate Biofilm and Adherence Formation in Klebsiella pneumoniae"

_microorganisms, 2025, doi:10.3390/microorganisms13112591_

Round 1

Reviewer 1 Report

Comments and Suggestions for Authors

The manuscript comfirmed that Fis acts as a transcriptional repressor of mrkA and mrkH,  negatively regulating the expression of type 3 fimbriae, biofilm formation, and adherence. Overall, the study is well-structured and sound; however, there are some issues that should be addressed to improve clarity and impact.

  1. The authors should further elaborate on the distribution and conservation of the gene of Fis in the genome of Klebsiella pneumoniaeto enhance the universality of these research findings.
  2. In the section of “2.9. Purification of Recombinant Fis-His₆ Protein”, How to express the homodimer of recombinant Fis protein, whose structure is crucial for its role in binding DNA as a regulator? The details should be described.
  3. For the result section of “3.3. Fis Suppresses Biofilm Formation and Host-Cell Adherence”, the author should provide the imaging data of biofilm experiments , such as the crystal violet staining picture of each well in Figure 3.
  4. Since the authors identified four potentialFis-binding motifs within the promoter region of mrkH based on the E. coli Fis consensus motif, the specific binding target site of Fis in KP could be verified by EMSA or other assays, ensuring that the conclusions would be more definitive than they currently are.

Author Response

The manuscript comfirmed that Fis acts as a transcriptional repressor of mrkA and mrkH,  negatively regulating the expression of type 3 fimbriae, biofilm formation, and adherence. Overall, the study is well-structured and sound; however, there are some issues that should be addressed to improve clarity and impact.

Response:

We sincerely thank the reviewer for their positive evaluation of our work and for the constructive comments that contributed to enhancing the clarity and impact of our manuscript. We have carefully addressed each comment, as detailed below. All additional text related to your suggestions is highlighted in green.

Comment 1.

The authors should further elaborate on the distribution and conservation of the gene of Fis in the genome of Klebsiella pneumoniae to enhance the universality of these research findings.

Response:
We appreciate this valuable suggestion. To address it, we have incorporated three paragraphs in the Introduction discussing the conservation and genomic distribution of the fis gene in K. pneumoniae and related Enterobacteriaceae. (New text added in Introduction, lines 80–98).

Comment 2.

In the section of “2.9. Purification of Recombinant Fis-His₆ Protein”, how to express the homodimer of recombinant Fis protein, whose structure is crucial for its role in binding DNA as a regulator? The details should be described.

Response:
We thank the reviewer for this valuable observation. We have expanded the Materials and Methods (Section 2.9) to include a detailed description of the renaturation and dimerization process of the recombinant Fis protein. After purification under denaturing conditions, the eluted Fis-His₆ fractions were subjected to stepwise dialysis to promote refolding and re-formation of the native homodimeric structure, which is essential for DNA-binding activity. This information has now been added to the revised manuscript (lines 301–310).

Comment 3.

For the result section of “3.3. Fis Suppresses Biofilm Formation and Host-Cell Adherence”, the author should provide the imaging data of biofilm experiments, such as the crystal violet staining picture of each well in Figure 3.

Response:
We fully agree that visual evidence strengthens our quantitative data. We have now included a representative image of the crystal-violet-stained wells, showing the biofilm biomass of WT, Δfis, and Δfis::fis strains, alongside the quantitative results. This image has been incorporated as Figure 3B, and the legend has been modified accordingly to indicate that the blank corresponds to TSB medium without inoculation.
(Revised Figure 3 and legend in Results section, lines 457–459).

Comment 4.

Since the authors identified four potential Fis-binding motifs within the promoter region of mrkH based on the E. coli Fis consensus motif, the specific binding target site of Fis in KP could be verified by EMSA or other assays, ensuring that the conclusions would be more definitive than they currently are.

Response:
We would like to extend our heartfelt thanks to the reviewer for their insightful and constructive suggestion. We completely agree that pinpointing the exact Fis-binding site within the mrkH promoter would significantly enhance our study. To clarify our experimental approach and rationale, we’d like to outline the sequence of our research.

Initially, we conducted EMSA assays (shown in Figure 4) using a probe that captures the entire regulatory regions of mrkH and mrkA. These experiments yielded a distinct and specific mobility shift in the presence of purified Fis protein, providing us with compelling evidence that Fis binds to the mrkH and mrkA promoters, thereby directly regulating these genes.

Following this promising demonstration of direct interaction, we proceeded with in silico analyses (Figure 5) aimed at identifying potential Fis-binding motifs within the mrkH and mrkA promoter regions. This bioinformatic analysis was designed to complement and enrich our EMSA findings by pinpointing sequence motifs that align with the known E. coli Fis consensus. The motifs we discovered are positioned in key functional areas of the mrkH and mrkA promoters, and notably, they do not overlap; however, one motif does partially intersect with the mrkH –35 box. This arrangement strongly indicates that Fis binding could impede promoter accessibility and transcriptional activity, affirming its role as a global architectural regulator, specifically as a transcriptional repressor in our study.

We fully acknowledge that determining the specific nucleotides bound by Fis would necessitate techniques such as site-directed mutagenesis or DNase I footprinting assays. However, we must note that these methods present significant technical challenges, as the predicted Fis-binding motifs are situated very close to the core promoter elements. Mutating these regions individually may inadvertently disrupt the promoter’s structural integrity or alter DNA curvature characteristics crucial for binding by nucleoid-associated proteins like Fis. Moreover, Fis typically engages with and bends DNA over extended regions, rather than at sharply defined sites, which could complicate the analysis of isolated mutations. Additionally, conducting DNase I footprinting assays is not currently viable in our lab due to the restrictions on radioisotopes, which we do not have the authorization to use.

Taking the reviewer’s valuable feedback into account, we have now included a clear statement in the Discussion section indicating that future research in our laboratory will focus on experimentally validating the in silico predictions through a detailed analysis of the Fis-binding motifs, aiming for a more definitive conclusion (lines 591-608).

We genuinely appreciate the reviewer’s keen observations, which we view as a fantastic opportunity for further advancing our research. We sincerely hope that our rationale for maintaining the current experimental framework, which we believe is scientifically robust and appropriate given the scope of this study and the technical considerations it entails, resonates. We are excited about the evidence presented, where the experimental EMSA data, paired with in silico motif analysis, creates a solid foundation supporting our conclusion that Fis directly regulates mrkH and mrkA and plays a significant role in controlling type 3 fimbriae expression in K. pneumoniae. Thank you once again for your kind and constructive input.

Reviewer 2 Report

Comments and Suggestions for Authors

This article examines the regulatory role of the fis gene in K. pneumoniae, an opportunistic pathogen recognized as a leading cause of nosocomial infections. Adhesion of pathogens to biotic and abiotic surfaces is a key factor in colonization and pathogenesis, making the article's relevance compelling.

The article's materials and methods are described in detail. The regulatory properties of Fis are confirmed by deletion and complementation approaches.

Overall, the work is very logically structured. It is possible that some insights from molecular genetic studies can be incorporated into the results.

Questions:

  1. What other pathogenicity factors might the Fis gene influence (capsule, siderophores, endotoxin)?
  2. Is it possible that Fis influences antibiotic resistance in Klebsiella?
  3. Is it possible that Fis influences bacterial biofilm formation across species in mixed communities? Some minor comments

Lines 99-100: Please indicate the manufacturer and country of the antibiotics used.

Insert statistical analysis into Figure 1. Was the growth of the strain with the Fis deletion different from the control?

Author Response

This article examines the regulatory role of the fis gene in K. pneumoniae, an opportunistic pathogen recognized as a leading cause of nosocomial infections. Adhesion of pathogens to biotic and abiotic surfaces is a key factor in colonization and pathogenesis, making the article's relevance compelling. The article's materials and methods are described in detail. The regulatory properties of Fis are confirmed by deletion and complementation approaches. Overall, the work is very logically structured. It is possible that some insights from molecular genetic studies can be incorporated into the results.

Response:

We sincerely thank the reviewer for the positive and encouraging comments regarding the structure, clarity, and relevance of our work. We have carefully considered the reviewer’s questions and suggestions, and our detailed responses are provided below. All additional text related to your suggestions is highlighted in yellow.

Comment 1.

What other pathogenicity factors might the Fis gene influence (capsule, siderophores, endotoxin)?

Response:

We appreciate this insightful question. As a global nucleoid-associated regulator, Fis can influence the transcription of multiple virulence-related genes in Enterobacteriaceae. Although our current study focused on the control of type 3 fimbriae, we have added a statement in the Discussion section acknowledging that Fis could potentially affect other virulence determinants, such as capsule production, siderophore biosynthesis, and LPS-associated genes. This represents an important avenue for future research (lines 643-649).

Comment 2.

Is it possible that Fis influences antibiotic resistance in Klebsiella?

Response:

We thank the reviewer for raising this important point. Although our study did not directly evaluate antibiotic resistance, the potential for Fis to influence antibiotic susceptibility cannot be excluded. In other bacteria, Fis has been linked to the regulation of efflux pump genes, membrane permeability, and stress responses, which can indirectly affect antibiotic resistance. Given these parallels, it is reasonable to suggest that Fis could similarly influence antibiotic resistance mechanisms in K. pneumoniae, either directly or through its global effects on membrane physiology. We have now included a statement in the Discussion highlighting this possibility as a future line of investigation (lines 650-658).

Comment 3.

Is it possible that Fis influences bacterial biofilm formation across species in mixed communities?

Response:

We appreciate the reviewer's intriguing question and broader ecological perspective. Fis is known to modulate biofilm formation by regulating surface structures and stress responses; however, its role in mixed-species biofilms has not yet been experimentally explored. Because Fis controls genes associated with adhesion and extracellular matrix production, it is reasonable to hypothesize that Fis could affect interspecies interactions within polymicrobial communities. We have added a sentence in the Discussion noting that Fis-dependent regulation might influence biofilm dynamics not only in K. pneumoniae monocultures but also in multispecies biofilms, a hypothesis that warrants future investigation (lines 659-665).

Minor comments

Lines 99–100: Please indicate the manufacturer and country of the antibiotics used.

Response:

We thank the reviewer for this observation. We have now included the manufacturer and country of origin for each antibiotic in the Materials and Methods section (line 118 in the revised version).

Insert statistical analysis into Figure 1. Was the growth of the strain with the fis deletion different from the control?

Response:

We appreciate the reviewer’s attention to this point. We have included statistical analysis for Figure 1 and clarified this in the legend. The growth curves of the wild-type and Δfis strains were significantly different under the tested conditions (*p<0.05, one-way ANOVA), indicating that the deletion of fis affects the general growth rate or fitness in rich medium.